# Dual-Satellite Lunar Global Navigation System Using Multi-Epoch Double-Differenced Pseudorange Observations

**Toshiki Tanaka [1,\*], Takuji Ebinuma [2] and Shinichi Nakasuka [1]**

[1] Department of Aeronautics and Astronautics, University of Tokyo, 7-3-1 Hongo, Bunkyo-ku, Tokyo 113-8656, Japan; nakasuka@space.t.u-tokyo.ac.jp

[2] Department of Astronautics and Aeronautics, Chubu University, 1200 Matsumoto-cho, Kasugai, Aichi 487-8501, Japan; ebinuma@isc.chubu.ac.jp

[\*] Correspondence: toshiki@nsat.t.u-tokyo.ac.jp

**Abstract:** In view of the upcoming missions to obtain resources from the lunar surface, it is essential to have highly-accurate navigation systems to locate surface vehicles in shadowed regions. In response, we propose a dual-satellite lunar navigation system that is based on a multi-epoch double-differenced pseudorange observations (MDPO) algorithm. We used multi-epoch observations in a new way that reduces the number of navigation satellites required. In addition, the double-differenced pseudorange is used in order to eliminate the bias effects of the satellite and user clocks that conventional dual-satellite navigation algorithms did not fully take into account. Furthermore, a pre-known lunar digital elevation model is used to reduce the number of observations. The theoretical behavior of the MDPO algorithm was confirmed by simulation and the results indicate that user position accuracy can be several tens of meters with 95% probability (2drms) within a one-minute observation.

**Keywords:** GNSS; navigation; microsatellite; nanosatellite; moon; lunar rover; interplanetary missions

## 1. Introduction

The estimation of a rover vehicle's position on the lunar surface is one of the key technologies for the successful operation of the rover, and it is also important for mapping resources and making scientific observations on the lunar surface. From an operational perspective, data concerning the position of the rover are vitally important in order to plan safe paths for the rover to take. From resource mapping and scientific observation perspectives, the position of the rover must be known in order to assign observed objects to proper locations on the lunar map. Some studies have reported that positions on the lunar surface must be known within an accuracy of 100 m to support both purposes [1].

It is well-known that valuable resources, including water ice and volatile compounds are located in the permanently shadowed regions (PSRs) on the lunar surface. Because of the lack of illumination by sunlight in the PSRs, navigation methods that use visual images, such as visual simultaneous localization and mapping (SLAM), will be constrained significantly if not completely invalidated. Therefore, some alternative navigation methodology is needed to enable long and efficient exploration of the PSRs. From another perspective, in order to reduce the risk associated with lunar exploration missions, the use of nano-rovers and/or micro-rovers is being discussed [2]. Since the locations of various resources are not known precisely, wide-range exploration by multiple small rovers must be conducted to identify the locations of resources precisely. Therefore, multiple-user navigation is urgently required. Furthermore, considering that the budget for early-stage lunar exploration is limited, we reason that a low-cost system using microsatellites has a great potential to accelerate lunar

missions. In summary, our target in this study is a low-cost micro-sized lunar navigation satellite system that can provide precise locations, i.e., within 100 m, for multiple small rovers.

To date, navigation technologies for the shadowed regions have been studied extensively. One study investigated the use of rovers with visual sensor-based navigation using a stereo camera with artificial light to explore the PSRs [3,4]. However, considering that a huge amount of power (calculated to be 864 W based on [3,4]) is required to provide sufficient light to illuminate a broad and continuous area, this approach is limited to large-scale rovers, i.e., rovers that weigh several hundred kilograms. Another study was conducted that considered the use of another type of visual sensor-based navigation, i.e., laser triangulation systems, on rovers to identify terrain profiles in the shadowed regions [3]. A laser triangulation system effectively uses its own light source by limiting the sensing area, and its power consumption can be reduced to a few watts by carefully selecting short distances, i.e., a few meters [3]. However, such applications have an inherent problem in that they cannot determine distances if the lunar surface is flat and repetitive and if there are no landmarks that can be used to assess distances [5]. Therefore, given the uncertainty of lunar terrains, there is significant risk associated with using visual sensor-based methods that have limited sensing ranges. Another study investigated a combination of the rover's inertia accelerometers and star tracker measurements for navigating rovers on the lunar surface [6]. In order to achieve 100-m position accuracy with this method, theoretically, 11.8-arcsecond user attitude determination is required, which is not feasible in the presence of sensor alignment errors. Another recent study introduced the idea of deploying orbiters in Halo orbits to establish a lunar global navigation satellite system (Lunar GNSS) analogous to Earth GNSS [7,8]. The Lunar GNSS proposed by these previous studies was based on time of arrival (TOA), i.e., pseudorange measurement, and it requires that at least four satellites be visible all the time, which inevitably requires a large number of satellites and, consequently, a large cost, but it only requires users to carry a passive ranging receiver. In an attempt to further reduce the costs of the system, some researchers have investigated the reduction of the number of satellites. Navigation technologies that use fewer than two satellites with a passive user receiver have been discussed extensively in the field of Earth GNSS applications [9]. One of these studies used angle of arrival (AOA) data and reduced the number of navigation satellites down to one [10]. However, this algorithm provides low position accuracy because a very small error in the AOA measurement results in a large error in the user position; i.e., a 1-degree error in the AOA for a distance of a few hundred km between the satellite and the user results in an error of a few kilometers in the position on the lunar surface. Another algorithm uses time difference of arrival (TDOA or single-differenced pseudorange) and/or frequency difference of arrival (FDOA, or single-differenced Doppler) to reduce the number of navigation satellites to a minimum of two [11]. Moreover, one study successfully showed that single-differenced Doppler using a static reference station, known as Law of Cosines (LOC), can provide a high-accuracy position with as little as one satellite on the lunar surface at the specified condition with several tens of minutes observation [12]. Furthermore, the authors of reference [12] also proposed an algorithm that uses a combination of range and Doppler measurements with a static reference station, known as Joint Doppler and Ranging (JDR), and achieved 3D positioning with as little as one satellite [13]. Basically, these previous studies [7–13] are based on the assumption that both the satellite and the user system or either the satellite system or the user system can provide a stable clock and/or a stable frequency without an offset (bias), and the estimation algorithms do not have to account for those errors. Such approaches also require the satellites and/or the user to carry a highly stable clock source, such as an atomic clock, in order to maintain a sufficiently small clock bias and frequency bias between bias estimations by ground segments (i.e., ground stations on the earth); otherwise the accuracy of the user position deteriorates immediately. In addition, the target of our study comprises a micro-sized satellite and rover systems whose power generation capability is limited by size and consequently, not compatible with the deep space atomic clock (DSAC). In this case, the best current clock technology that is compatible with the micro-sized satellite is the Chip Scale Atomic Clock (CSAC). As reported in [14], while CSAC can suppress the frequency instability

of the clock down to about 1 ppb for 24 h, CSAC incurs several tens to hundreds of meters of error in pseudorange observation after 24 h, which further increases over time. As a result, using CSAC inevitably requires pseudorange-based navigation systems to conduct frequent estimations of the satellite and/or user clock bias using earth ground stations, which is very challenging in Lunar GNSS because of the limitation of the availability and number of earth ground stations that are capable of Earth–Moon distance communication. Another approach to eliminating the biases of the satellite and user clocks at the same time is to use two-way ranging between the user and the satellite [15]. However, this requires an active ranging operation between the satellite segment and the user segment (the user must send a radio signal to the satellite, and the satellite has to send the received radio signals back to each user separately), and this consequently imposes an extra burden and cost on the user segment as well as on the satellite side. In order to reduce the total cost of the system, including the user segment, a method that uses passive ranging is ideal, especially for the multiple rover missions that will occur in the future. Contrary to these previous studies, our research uses multi-epoch double-differenced pseudorange observations (MDPO), which is a passive system using a static reference station, and it works while accounting for the instability of both the satellite and user clocks. The comparison of the proposed method with other conventional methods is summarized in Table 1.

**Table 1.** Benchmark of navigation systems for lunar shadowed region exploration.

| Method | User (Rover) Segment Burden | Space (Satellite) Segment Burden | Ground Segment Burden |
|---|---|---|---|
| Visual Sensor-based Navigation | Visual Sensor-based navigation does not work when the lunar surface is flat with no landmarks. | - | - |
| Accelerometers and Star Tracker Navigation | Sensor alignment precision becomes outrageous to achieve high position accuracy. | - | - |
| Lunar Global Navigation Satellite Systems using TOA | Use a passive ranging receiver. | At least four satellites in view with a stable satellite clock are required. | Frequent satellite clock bias estimation by the ground segment is required. |
| Single Satellite AOA Navigation | Use a passive ranging receiver. User position accuracy is very sensitive to AOA error. | Single satellite in view with a stable satellite clock is required. | Frequent satellite clock bias estimation by the ground segment is required. |
| Dual Satellite TDOA/FDOA Navigation | Use a passive ranging and/or Doppler receiver. | Two satellites in view with a stable satellite clock and/or frequency are required. | Frequent satellite clock bias estimation by the ground segment is required. |
| Law of Cosines | Use a passive Doppler receiver with a static reference station. The frequency of the receiver must be stable. | Single satellite in view is required, with no need for a stable satellite frequency. | No need for frequent satellite clock bias estimation by the ground segment. |
| Joint Doppler and Ranging (single satellite case) | Use a passive ranging and Doppler receiver with a static reference station. The clock and frequency of the receiver must be stable or must be compensated by two-way ranging. | Single satellite in view is required, with no need for a stable satellite clock. | No need for frequent satellite clock bias estimation by the ground segment. |

**Table 1.** *Cont.*

| Method | User (Rover) Segment Burden | Space (Satellite) Segment Burden | Ground Segment Burden |
|---|---|---|---|
| Two-way Ranging based Navigation | Active ranging between the satellite and user is required. | Two satellites in view are required, with no need for a stable satellite clock. | No need for frequent satellite clock bias estimation by the ground segment. |
| Dual Satellite MDPO Navigation (This research) | Use a passive ranging receiver with a static reference station. | Two satellites in view are required, with no need for a stable satellite clock. | No need for frequent satellite clock bias estimation by the ground segment. |

In this study, we target mobile applications such as lunar rovers as users, which requires observation periods to be small. Since a method using only Doppler observation needs several tens of minutes of observation to provide a high user position accuracy [12], pseudorange observation data must be used. Besides this, regarding biases of the satellite and user clocks, methodologies using TOA or TDOA (pseudorange or single-differenced pseudorange) are insufficient because they were not designed to cope with both the satellite and user clock biases at the same time. Instead, in this study, double-differenced pseudorange was used to remove the bias of both the satellite and user clocks from the estimation (note that the double-differenced pseudorange is explained further in the following section). Moreover, our proposed method does not use frequency observation and uses the double-differenced pseudorange only, which can contribute to making the pseudorange receiver hardware design as simple as possible. Furthermore, in order to reduce the number of satellites, we introduce multi-epoch observations that use pseudorange measurements from multiple epochs (Figure 1).

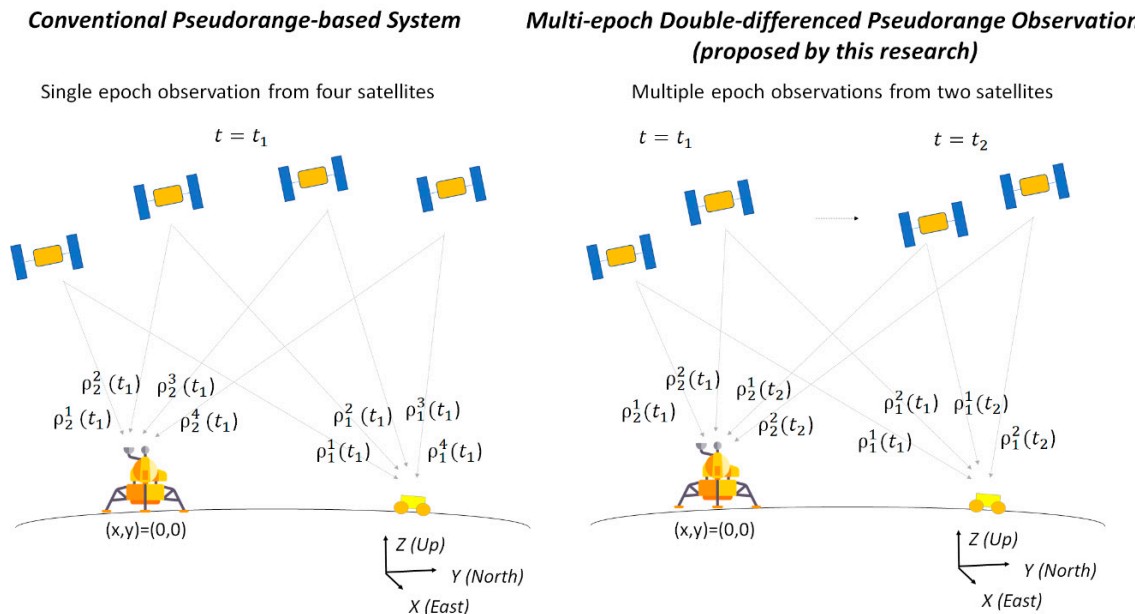

**Figure 1.** Overview of the MDPO concept in comparison with a conventional pseudorange-based method.

This paper consists of the following sections. In Section 2, the algorithm of MDPO is formulated and the expected accuracy of user position is discussed. In Section 3, the theoretical behavior of MDPO is confirmed by numerical simulation, along with achievable user position accuracy. Section 4 provides our conclusions.

## 2. Algorithm

### 2.1. Multi-Epoch Double-Differenced Pseudorange Observations (MDPO) Algorithm

In a pseudorange-based algorithm, the pseudorange ($\rho$) observation between one user and one satellite is presented by the following equation:

$$\rho_R^S(t_i) = r_R^S(t_i) + c\big(d\tau_R(t_i) - dT^S\big(t_i^s\big)\big) + \omega_R^S(t_i) \tag{1}$$

$$r_R^S(t_i) = \sqrt{\big(x^S\big(t_i^s\big) - x_R(t_i)\big)^2 + \big(y^S\big(t_i^s\big) - y_R(t_i)\big)^2 + \big(z^S\big(t_i^s\big) - z_R(t_i)\big)^2} \tag{2}$$

where $\big(x^S\big(t_i^s\big), y^S\big(t_i^s\big), z^S\big(t_i^s\big)\big)$ is the satellite position at the time of signal transmission $t_i^s$, $(x_R(t_i), y_R(t_i), z_R(t_i))$ is the user position at the time of signal reception $t_i$, $c$ is the speed of light, $d\tau_R$ is user clock bias, $dT^S$ is satellite clock bias, and $\omega_R^S$ is receiver observation error. In this study, we assume that receiver observation error $\omega_R^S$ follows a white Gaussian distribution. The coordinate frame of the satellite position and user position is based on a topocentric frame that is a Moon-fixed frame, with the origin of the frame being at the user position: i.e., the *x*-axis points local east, the y points local north and the *z*-axis points local up (East-North-Up). The equations are formulated using the relative position between the satellite and the user, and both the satellite and user positions have a constant rotational offset with respect to the Moon-centered inertial frame.

A method called double difference is used to remove both the satellite and user clock biases from estimation parameters, by subtracting four pseudorange observations between two users (user1, user2) and two satellites (satellite1, satellite2), as shown in Equations (3)–(7):

$$\rho_1^1(t_i) = r_1^1(t_i) + c\big(d\tau_1(t_i) - dT^1\big(t_i^1\big)\big) + \omega_1^1(t_i) \tag{3}$$

$$\rho_1^2(t_i) = r_1^2(t_i) + c\big(d\tau_1(t_i) - dT^2\big(t_i^2\big)\big) + \omega_1^2(t_i) \tag{4}$$

$$\rho_2^1(t_i) = r_2^1(t_i) + c\big(d\tau_2(t_i) - dT^1\big(t_i^1\big)\big) + \omega_2^1(t_i) \tag{5}$$

$$\rho_2^2(t_i) = r_2^2(t_i) + c\big(d\tau_2(t_i) - dT^2\big(t_i^2\big)\big) + \omega_2^2(t_i) \tag{6}$$

$$\Delta\nabla\rho(t_i) = \rho_1^1(t_i) - \rho_1^2(t_i) - \big(\rho_2^1(t_i) - \rho_2^2(t_i)\big) = r_1^1(t_i) - r_1^2(t_i) - \big(r_2^1(t_i) - r_2^2(t_i)\big) + \\ \omega_1^1(t_i) - \omega_1^2(t_i) - \big(\omega_2^1(t_i) - \omega_2^2(t_i)\big) = \Delta\nabla r(t_i) - \Delta\nabla\omega(t_i) \tag{7}$$

where $\Delta\nabla$ denotes double difference. In the double difference method, user2 is used as a reference station whose position is fixed and known, and the position of user1 is estimated in relation to the position of user2; i.e., user2's position is referenced as the origin of navigation (0,0,0). In a lunar navigation system, the lander can be used as a reference station (user2), and its geodetic position is used as the origin of navigation. Note that the geodetic position of the lander must be obtained in advance of the start of the rover navigation by other means, such as identification by satellite image. Hereafter, the rover corresponds to user1 and the lander corresponds to user2.

In the MDPO algorithm, multiple double-differenced pseudorange observations, i.e., $\Delta\nabla\rho(t_k), \ldots, \Delta\nabla\rho(t_{k+N-1})$, are obtained from multiple epochs, i.e., $t_k - t_{k+N-1}$, where N is the number of observed epochs, and k is the epoch number at which the estimation starts. Note that the rover position must be fixed during multi-epoch observations taken in place in order to keep the number of estimation parameters less than the number of observation equations. Otherwise, the rover position cannot be identified deterministically by the MDPO algorithm and the rover position accuracy changes depending on the quality of other navigation information used during multi-epoch observations. Hereafter, $(x_R(t_k), y_R(t_k), z_R(t_k))$ represents a fixed rover position during $t_k - t_{k+N-1}$.

The standard approach to solving nonlinear systems, such as Equation (7), is known as the Newton–Raphson method, which is a general iterative method that uses linear regression to find the

root of a function. The idea is to start with rough estimates of the rover position and refine them in stages so that the estimates fit the observations better. First, the range is calculated on an initial estimated value of the rover position $X_R^0(t_k) = \left( x_R{}^0(t_k),\ y_R{}^0(t_k), z_R{}^0(t_k) \right)$:

$$r_R^{s\ 0}(t_i) = \sqrt{\left( x^S\!\left(t_i^s\right) - x_R{}^0(t_k) \right)^2 + \left( y^S\!\left(t_i^s\right) - y_R{}^0(t_k) \right)^2 + \left( z^S\!\left(t_i^s\right) - z_R{}^0(t_k) \right)^2} \tag{8}$$

$$i = k, \ldots, k + N - 1$$

Practically, the previous position of the rover can be used as an initial guess, i.e., $X_R^0(t_k) = X_R(t_{k-1})$. Then, the double-differenced range on the initial estimated value of the rover position is calculated as

$$\Delta\nabla r^0(t_i) = r_1^{1\ 0}(t_i) - r_1^{2\ 0}(t_i) - \left( r_2^{1\ 0}(t_i) - r_2^{2\ 0}(t_i) \right) \tag{9}$$

We also define a new parameter $R$:

$$R(t_i) = \Delta\nabla\rho(t_i) - \Delta\nabla r^0(t_i) \tag{10}$$

where $R$ is the difference between the measured double-differenced pseudorange value and the calculated double-differenced range. By substituting Equation (10) into Equation (7), the difference between the true double-differenced range and the calculated double-differenced range, i.e., $\Delta\nabla r(t_i) - \Delta\nabla r^0(t_i)$, can be described:

$$\Delta\nabla r(t_i) - \Delta\nabla r^0(t_i) = R(t_i) + \Delta\nabla\omega(t_i) \tag{11}$$

On the other hand, the residual error between the true double-differenced range and the calculated double-differenced range can be written using a Taylor series approximation as:

$$\Delta\nabla r(t_i) - \Delta\nabla r^0(t_i) = \frac{\partial\Delta\nabla r(t_i)}{\partial x}(\Delta x) + \frac{\partial\Delta\nabla r(t_i)}{\partial y}(\Delta y) + \frac{\partial\Delta\nabla r(t_i)}{\partial z}(\Delta z) \tag{12}$$

$$\Delta x = x_R(t_k) - x_R{}^0(t_k) \tag{13}$$

$$\Delta y = y_R(t_k) - y_R{}^0(t_k) \tag{14}$$

$$\Delta z = z_R(t_k) - z_R{}^0(t_k) \tag{15}$$

By substituting Equation (11) into (12), the following equation is obtained:

$$R(t_i) = \frac{\partial\Delta\nabla r(t_i)}{\partial x}(\Delta x) + \frac{\partial\Delta\nabla r(t_i)}{\partial y}(\Delta y) + \frac{\partial\Delta\nabla r(t_i)}{\partial z}(\Delta z) - \Delta\nabla\omega(t_i) \tag{16}$$

Equation (16) for multiple epochs $t_k - t_{k+N-1}$ can be written at once using the following matrix:

$$R = G\Delta X + w \tag{17}$$

$$R = \begin{bmatrix} R(t_k) & \cdots & R(t_{k+N-1}) \end{bmatrix}^T \tag{18}$$

$$\Delta X = [\Delta x, \Delta y, \Delta z] \tag{19}$$

$$w = \begin{bmatrix} -\Delta\nabla\omega(t_k) & \cdots & -\Delta\nabla\omega(t_{k+N-1}) \end{bmatrix}^T \tag{20}$$

$$G = \begin{bmatrix} \dfrac{\partial\Delta\nabla r(t_k)}{\partial x} & \dfrac{\partial\Delta\nabla r(t_k)}{\partial y} & \dfrac{\partial\Delta\nabla r(t_k)}{\partial z} \\ \vdots & \vdots & \vdots \\ \dfrac{\partial\Delta\nabla r(t_{k+N-1})}{\partial x} & \dfrac{\partial\Delta\nabla r(t_{k+N-1})}{\partial y} & \dfrac{\partial\Delta\nabla r(t_{k+N-1})}{\partial z} \end{bmatrix} \tag{21}$$

where $G$ is known as an observation matrix. By solving the least-square problem that minimizes the residual error $|R - G\Delta X|$, an estimated value of $\Delta X$, defined as $\hat{\Delta X}$, is obtained:

$$\hat{\Delta X} = \left(G^T G\right)^{-1} G^T R \tag{22}$$

Then, a new estimated value $X_R^1(t_k) = \left(x_R{}^1(t_k), y_R{}^1(t_k), z_R{}^1(t_k)\right)$ is given by Equation (23), which provides a better fit to the observation.

$$X_R^1(t_k) = X_R^0(t_k) + \hat{\Delta X} \tag{23}$$

This estimation process continues $(X_R^1, X_R^2)$ until the number of iterations reaches the designed value, i.e., $n$, and then final estimated value $X_R^n(t_k)$ is acquired. To estimate the rover's three-dimensional position $X_R = (x_R, y_R, z_R)$, the number of multi-epoch observations must be larger than 3 (N > 3).

Next, we characterize the quality of the estimates. We can write an expression for the error as

$$\hat{\Delta X} - \Delta X = \left(G^T G\right)^{-1} G^T (G\Delta X + w) - \Delta X = \left(G^T G\right)^{-1} G^T w \tag{24}$$

Suppose that $w$ follows a white Gaussian distribution that has a mean value of zero and covariance matrix C: the covariance of $\hat{\Delta X} - \Delta X$, defined as $P$, is given by

$$P = \left(G^T G\right)^{-1} G^T C G \left(G^T G\right)^{-1} \tag{25}$$

The expression becomes much simpler if the components of $w$, i.e., $-\Delta\nabla\omega(t_i)$, are uncorrelated and have an identical variance, i.e., $C = \sigma_{\Delta\nabla\omega}^2 I$:

$$P = \sigma_{\Delta\nabla\omega}^2 \left(G^T G\right)^{-1} \tag{26}$$

where $\sigma_{\Delta\nabla\omega}^2$ is the variance of double-differenced receiver observation errors. In GNSS terminology, $\left(G^T G\right)^{-1}$ is known as the dilution of precision (DOP) matrix, which is used to specify error propagation as a mathematical effect of navigation satellite geometry on positional measurement precision. We define the DOP matrix and its elements $\sigma_{DOP}$ as

$$DOP = \begin{bmatrix} \sigma_{DOP\ 11} & \cdots & \sigma_{DOP\ 1N} \\ \vdots & \ddots & \vdots \\ \sigma_{DOP\ N1} & \cdots & \sigma_{DOP\ NN} \end{bmatrix} = \left(G^T G\right)^{-1} \tag{27}$$

where $\sigma_{DOP}$ is elements of $DOP$. By substituting Equation (27) into Equation (26), theoretically, the achievable rover position error, i.e., $\hat{\Delta X} - \Delta X$, at a time of $t_k$ is given by

$$UPE(t_k) = \left|\hat{\Delta X}(t_k) - \Delta X(t_k)\right| = \sqrt{\sum_{j=1}^{N} \left(\sigma_{DOP\ jj}\right)^2} \times \sigma_{\Delta\nabla\omega} \tag{28}$$

where $UPE$ represents the distance between the rover's true position and an estimated rover position. We define GDOP as

$$GDOP = \sqrt{\sum_{j=1}^{N} \left(\sigma_{DOP\ jj}\right)^2} \tag{29}$$

Then Equation (28) is written as

$$UPE(t_k) = GDOP \times \sigma_{\Delta\nabla\omega} \tag{30}$$

As mentioned in the previous section, we assume receiver observation errors follow a normal distribution with a zero mean (i.e., Gaussian white noise). As such, UPE also follows a 1-D Gaussian distribution, and 95 percent of it lies inside the interval from $-2s$ to $+2s$, where $s$ is the standard deviation. As a performance index, this research uses 2drms ($2s$), which is commonly used in two-dimensional position estimation problems. Furthermore, in the MDPO algorithm, the UPE value, as well as the GDOP value, changes over time, so an indicator that represents the overall UPE over the course of the mission time is needed. For this purpose, Total UPE is newly defined, along with Total GDOP, as below:

$$Total\ UPE = \sqrt{\frac{1}{m}\sum_{}^{m} UPE(t_k)} = Total\ GDOP \times \sigma_{\Delta\nabla\omega} \tag{31}$$

$$Total\ GDOP = \sqrt{\frac{1}{m}\sum_{}^{m} GDOP} \tag{32}$$

where m is the number of MDPO estimations over the course of the mission time. Note that $\sigma_{\Delta\nabla\omega}$ is independent of time and can be excluded from the square root without losing generality. As seen in Equation (31), Total UPE also follows a 1-D Gaussian distribution. Therefore, 2drms can also be used as a performance index for Total UPE.

### 2.2. Two-Dimentional MDPO Algorithm Using a Pre-Known User Altitude

It is known that when a user altitude $z_R$ is known by other means, $\Delta z$ in Equation (12) becomes zero and can be eliminated [16]. Accordingly, the z-spatial distribution $\frac{\partial \Delta\nabla r}{\partial z}$ can be removed from $G$ in Equations (17)–(21):

$$R = G\Delta X + w \tag{33}$$

$$R = \begin{bmatrix} R(t_k) & \cdots & R(t_{k+N-1}) \end{bmatrix}^T \tag{34}$$

$$\Delta X = [\Delta x, \Delta y] \tag{35}$$

$$w = \begin{bmatrix} -\Delta\nabla\omega(t_k) & \cdots & -\Delta\nabla\omega(t_{k+N-1}) \end{bmatrix}^T \tag{36}$$

$$G = \begin{bmatrix} \frac{\partial\Delta\nabla r(t_k)}{\partial x} & \frac{\partial\Delta\nabla r(t_k)}{\partial y} \\ \vdots & \vdots \\ \frac{\partial\Delta\nabla r(t_{k+N-1})}{\partial x} & \frac{\partial\Delta\nabla r(t_{k+N-1})}{\partial y} \end{bmatrix} \tag{37}$$

We call this method two-dimensional (2D) MDPO. This helps the MDPO algorithm achieve a smaller GDOP value in the same or a shorter observation period and, as a result, provide a better user position accuracy compared with three-dimensional estimation. Therefore, in this study, only the 2D MDPO algorithm is used hereafter. In 2D MDPO, the number of multi-epoch observations can be reduced to as low as 2 (N = 2).

In a lunar navigation problem, rover altitude $z_R$ can be pre-estimated using a lunar digital elevation model (DEM) [17,18]. As shown in Equation (38), the DEM is a function of longitude and latitude, which are not known at the start. The estimation of sequences proceeds in the following sequence: First, $X_R^0(t_k)$ is estimated using the rover position before its relocation, i.e., $X_R(t_{k-1}) = (x_R(t_{k-1}), y_R(t_{k-1}), z_R(t_{k-1}))$. Then, a new estimated rover position, i.e., $X_R^1(t_k)$, is estimated as $(x_R^1(t_k), y_R^1(t_k), z_R(t_{k-1}))$ by Equation (23). Note that $z_R$ is not updated at this moment. After that, the altitude of the rover is updated to $z_R^1(t_k)$ using $x_R^1(t_k)$ and $y_R^1(t_k)$ by Equation (38). The calculation continues until the number of iterations reaches the designed value, i.e., $n$.

$$z_R^i(t_k) = z_{R\ DEM}\left(x_R^i(t_k), y_R^i(t_k)\right) \tag{38}$$

Here, $z_{R\,DEM}$ is a lunar DEM model that is a function of latitude and longitude. Note that, according to Equation (38), when $z_R$ changes along with $x_R$ and $y_R$, errors in the X-Y position induces errors in the Z position, which ultimately induces errors in estimated $x_R$ and $y_R$ according to Equation (12), and as a result, Total UPE deteriorates stochastically. In our research, we do not apply the case in which the rover altitude changes too rapidly, such as the rover dropping off the cliff or roving on steep slopes. In that case, Total UPE will not deteriorate too significantly, which was confirmed by the simulations in the following section.

*2.3. Other Systematic Errors*

In an actual situation, with the presence of other systematic errors shown in this section, the discussed achievable Total UPE in Equation (31) will increase. In this section, the theoretical background of systematic errors as well as their impact on UPE is discussed. As the impact of such errors on UPE cannot be predicted analytically, we used a numerical simulation, reported in the following section, to quantitatively determine impact.

2.3.1. Satellite Orbit Determination Error

In the algorithm equations, the pseudorange $\rho$ is calculated on the basis of pre-estimated satellite positions $X^s = \left(x^S, y^S, z^S\right)$. In an actual situation, satellite orbit determination is not perfect, and pre-estimation of the satellite position entails some error relative to true positions ($\Delta X^S_{sat\,OD}$). According to a general satellite orbit determination process, the error is decomposed along with the satellite velocity direction (Along), satellite zenith direction (Radial), and cross-track direction (Cross). In this simulation, orbit determination error is defined along with the Along, Radial, and Cross directions and then converted into a user frame:

$$\Delta X^S_{sat\,OD}(t_i) = T \times \left(\Delta Along(t_i), \Delta Radial(t_i), \Delta Cross(t_i)\right) \tag{39}$$

where $T$ is a coordinate transformation matrix from Along, Radial, and Cross to a topocentric frame. The definition of the topocentric frame is explained in the previous chapter. In multilateration theory, only satellite orbit determination error in the line-of-sight direction (rover to satellite) matters, and other directions have almost no impact on rover position error. In the MDPO algorithm, line-of-sight direction error is eliminated by the Double Difference method, along with the satellite, the rover, and the lander clock biases. Hence, basically, satellite orbit determination error has no impact on the rover position error in the MDPO algorithm.

2.3.2. Time Tag Error

In the estimation process of the satellite position at a given time, the time tag of the receiver is used to propagate estimated satellite positions. In common GNSS systems, the receiver time tag is calibrated by a satellite clock via a navigation message. However, there is ambiguity in the signal traveling between satellites and the rover. As a result, the receiver time tag entails continuous bias error. As such, an estimated satellite position $X^s = \left(x^S, y^S, z^S\right)$ is deteriorated by the receiver clock bias $d\tau_R(t_i)$, and has some error relative to the true positions ($\Delta X^S_{time\,tag}$), such as

$$\Delta X^S_{time\,tag}(t_i) = \left(V_{x_R^S}(t_i), V_{y_R^S}(t_i), V_{z_R^S}(t_i)\right) \times d\tau_R(t_i) \tag{40}$$

where $\left(V_{x_R^S}, V_{y_R^S}, V_{z_R^S}\right)$ is a pre-estimated satellite relative velocity in a topocentric frame. Basically, satellite position error induced by a time tag error is eliminated from the estimation by the double difference method in a manner similar to the way in which satellite orbit determination error is removed.

### 2.3.3. DEM Information Error

As reported in [17,18], current lunar DEM information is developed from remote-sensing data and, as a result, is not perfect. Therefore, the DEM error $\Delta z_{R\ DEM}$ defined in Equation (41), which is the difference between the true rover vertical position $z_{R\ true}$ and a pre-given rover vertical position $z_{R\ DEM}$, leads to position estimation error in the X–Y plane $(x_R, y_R)$. The impact of DEM model error on X–Y position estimation accuracy appears stochastically, and its value changes depending on satellite position and velocity in relation to the rover and lander position.

$$\Delta z_{R\ DEM} = z_{R\ true} - z_{R\ DEM} \tag{41}$$

### 2.3.4. Other System Errors

In the general context of navigation satellite systems, other system errors need to be considered such as ionospheric delay, tropospheric delay, antenna phase characteristics, and multi-pass. However, such errors are negligible or not detrimental to rover position estimation in lunar surface navigation systems. Ionospheric delay and tropospheric delay are deemed negligible. Antenna phase characteristics appear in the same way and are almost negligible. Multi-pass is much smaller than earth's surface because there are fewer high objects in the surroundings. Therefore, these errors can be deemed ignorable and were not considered in this research.

### 2.4. Design Parameters

The spatial position of two satellites is one of the most important design parameters that directly impact the rover position accuracy. It is known that in order to acquire an accurate user position, a small DOP value is required, and accordingly, the distance between two satellites has to be large. In comparison, in order to keep both satellites in the rover's view for a long time, a short distance between two satellites is preferable. As a result, these two requirements conflict with each other, and both impacts must be carefully considered to find the best compromise point in the satellite trajectory selection. Figure 2 shows availability, which is the percentage of time at which both satellites are in the rover's view to total mission time, and the Total GDOP value at several orbit conditions: circular orbits with four different satellite altitudes (300 km, 600 km, 900 km, 2100 km) and five different orbital phase differences $\Delta\Omega$ between two satellites (5 deg, 15 deg, 25 deg, 35 deg). Rover/lander positions were fixed to the south-pole (−90 deg, 90 deg), and satellite orbital inclination was fixed to 110 deg without losing the generality of the discussion.

As seen in Figure 2, availability and Total GDOP have a negative correlation. At the same time, there are some good compromise points, such as "altitude 300 km/phase difference 15 deg," where both availability and Total GDOP have moderate values. The value of availability is also an important factor to consider for rover operation. In the case of two low lunar orbiters, the value of availability is limited up to around 15%. However, we think that this value is compatible with a mission that requires a higher availability value for long-range exploration: essentially, the rover can rely on its inertial navigation system (INS) when pseudorange-based navigation is not available and can retrieve the precise position once the pseudorange-based navigation is back in service. In order to keep the position error within 100 m all the time, the design margin should be considered in a way such that the navigation accuracy of the dual-satellite pseudorange navigation has a sufficient margin to 100 m, which is used to compensate for the position error induced by INS-based navigation during the time that dual-satellite pseudorange navigation is not available.

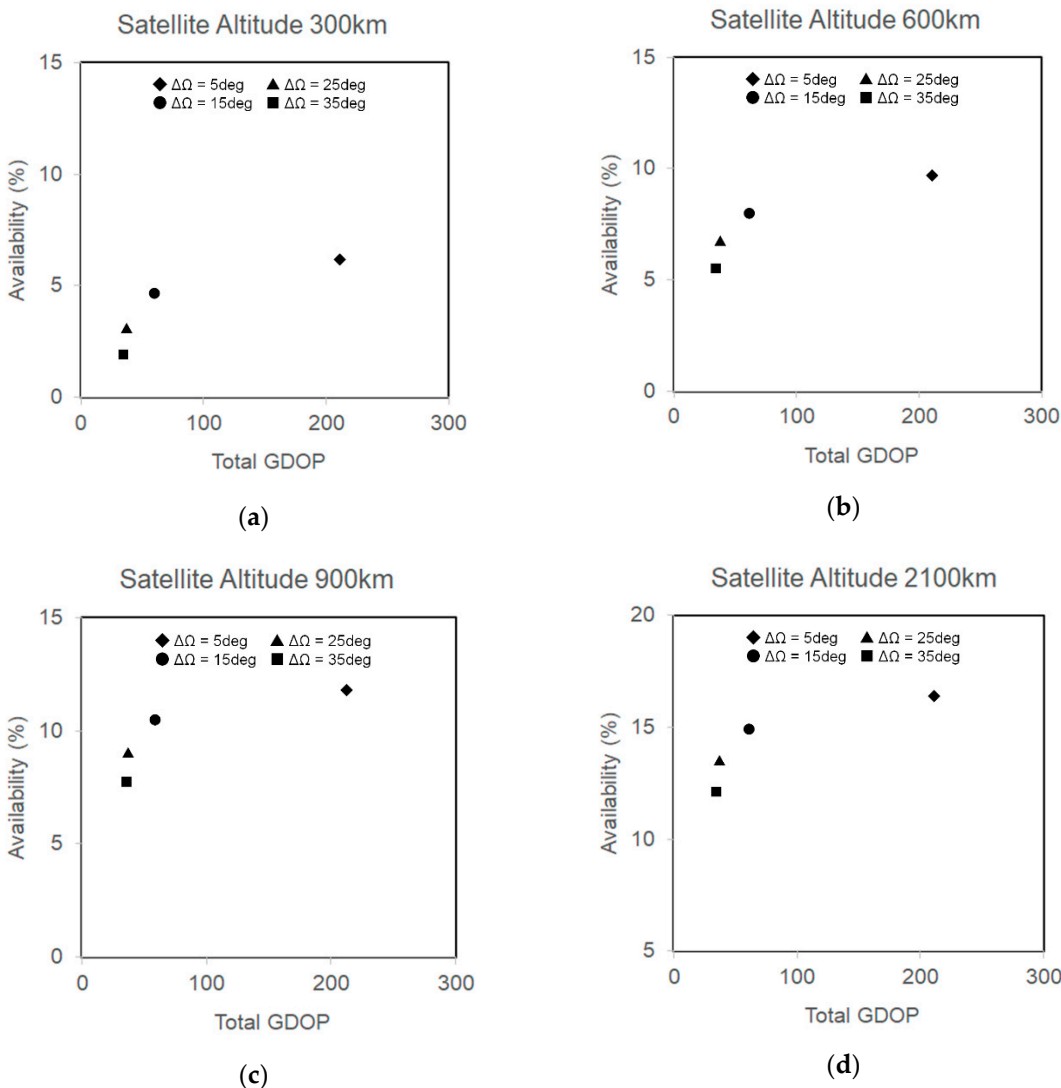

**Figure 2.** Availability and Total GDOP under different orbital conditions: circular orbits with four different satellite altitudes (300 km (**a**), 600 km (**b**), 900 km (**c**), 2100 km (**d**)) and five different orbital phase differences $\Delta\Omega$ between two satellites (5 deg, 15 deg, 25 deg, 35 deg). Rover/lander position were fixed to the south-pole (−90 deg, 90 deg) and satellite orbital inclination was fixed to 110 deg.

Another consideration is the required power for RF communication between satellites and the rover/lander. In general, the higher the orbit altitudes, the more power that is required by RF communication, and the free-space path loss increases proportionally to the square of the distance between the satellites and the rover/lander. In order to reduce the system burden for RF power on the satellite side as well as on the rover and lander side, a lower orbit, such as 300 km, is preferable.

Furthermore, for a long-period mission, orbit perturbation should also be considered. As a result of cis-lunar perturbation, some important orbital parameters, such as altitude and phase difference, are subject to change over time. In this study, the main focus is on algorithm verification and performance evaluation under specific conditions. Therefore, the selection of satellite trajectories used in the next section was not optimal, although it was good enough to maintain the phase difference between 15.0 deg and 17.5 deg over the course of the simulation period. Additionally, it is well-known that there are several stable orbit families that repeat ground tracks on the Moon, known as repeat ground track (RGT) orbits. For example, Ruam P. Russel successfully proved that stable or near-stable families of solutions exist for a full range of average inclinations and altitudes, making them suitable for

long-lifetime parking applications [19]. An optimal orbit should be chosen according to the specification of each mission.

## 3. Simulation

In order to assess user position accuracy in the presence of the systematic errors discussed in the previous chapter, we developed a numerical simulation model.

### 3.1. Simulation Overview

Figure 3 provides an overview of the simulation system. First, a rover trajectory in the X–Y direction, i.e., a time-series dataset of $x_R$ and $y_R$, was created, and then a rover position in the Z direction, i.e., $z_R$, was also created using lunar DEM data $z_{R\ DEM}$. Then, by adding DEM error ($\Delta z_{R\ DEM}$) to a created rover trajectory, the true rover position $X_{R\ true}$ was developed. For lunar DEM data, we used [20], which is 5-m resolution DEM data for latitude from −87.5 deg to −90 deg. The DEM error dataset, i.e., $\Delta z_{R\ DEM}$, was prepared at a 1-m grid interval. In other words, the DEM data change every 5-m grid, while DEM error data change every 1-m grid. The true rover altitude, i.e., the z-component of $X_{R\ true}$, is estimated using the DEM value and DEM error value of the closest grid point from its horizontal location respectively: e.g., if the rover is horizontally located at $(x_R, y_R) = (11.3\ [m], 3.5\ [m])$, it refers to the DEM data of the point $(x_R, y_R) = (10.0\ [m], 5.0\ [m])$ and the DEM error data of the point $(x_R, y_R) = (11.0\ [m], 3.0\ [m])$ to calculate the true rover altitude. Next, the true satellite trajectory $X_{true}^S$ was prepared separately. A precise cis-lunar dynamics model takes into account the gravity models of the Moon to degree 40 as well as the gravity from the Earth and the Sun, which was used to generate satellite trajectory data in the topocentric frame, whose origin is at the lander position. The true range was calculated using the true satellite, rover and lander positions while taking into account the Moon precession during the signal traveling time between the satellites and rover/lander. Then, by adding receiver observation errors to a true range value, the pseudorange observation $\rho_R^S(t_i)$ was prepared. Also, by adding Satellite orbit determination error and Time tag error to the satellite's true position $X_{true}^S$, the observed satellite position $X_{ob}^S$ was prepared. The 2D MDPO algorithm uses the pseudorange observation $\rho_R^S(t_i)$, observed satellite position $X_{ob}^S$ and lunar DEM data $z_{R\ DEM}$, and an estimated rover position $X_{R\ est}$ was calculated over the course of the simulation period. Finally, the true rover position $X_{R\ true}$ and the estimated rover position $X_{R\ est}$ were compared to evaluate the algorithm estimation accuracy.

Table 2 summarizes the general parameters used in the simulation. The total simulation period was set to 15,000 min assuming a two-week-long mission. Range measurement resolution at the user pseudorange receiver was set to 0.4 m assuming a typical space GNSS receiver specification with a safety margin. The initial rover position and lander position were set to (−90 deg, 90 deg) assuming a south-pole mission. The rover trajectory was created dynamically by changing the rover position after each MDPO estimation according to the defined traveling distance and the random heading direction specified in Table 2. Two-dimensional MDPO requires pseudorange observations from two epochs, and the interval of pseudorange observations was set to 0.5 min. Hence, it takes 1.0 min for the 2D MDPO algorithm to estimate the rover position. The rover position was fixed during the MDPO estimation for 1.0 min, and then the rover position was changed in the following 0.5 min and then stopped for 1.0 min for another MDPO estimation, which continued over the course of the simulation period. In addition, the rover moved only when both orbiters were in view. Initial satellite orbits were selected, according to the discussion in Section 2.3, to be a 110 deg–300 km (inclination–altitude) orbit with a 15–deg phase difference, as shown in Table 3.

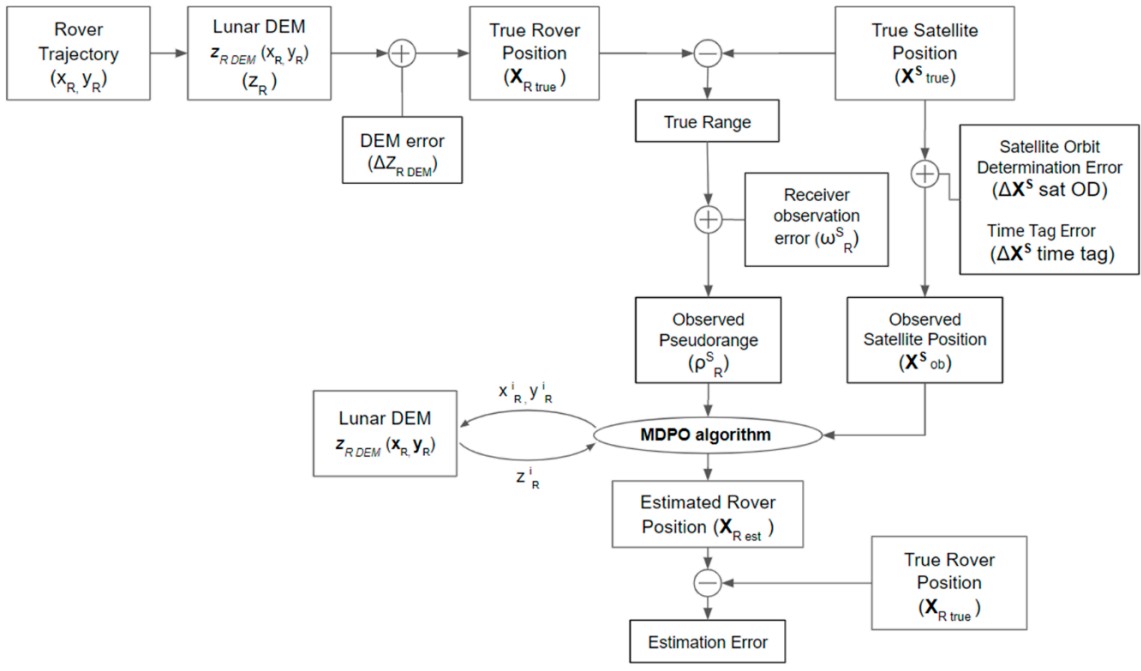

**Figure 3.** Simulation overview.

**Table 2.** Simulation parameters.

| Items | Value | Unit | Remarks |
|---|---|---|---|
| Simulation Period | 15,000 | min | Approximately two weeks in Earth time |
| Range measurement resolution of the user pseudorange receivers | 0.4 | m | Minimum observable range by the rover and lander receivers |
| Latitude of Initial Rover/Lander Position | −90 | deg | |
| Longitude of Initial Rover/Lander Position | 90 | deg | |
| Interval of pseudorange observations | 0.5 | min | Total observation period of one MDPO estimation is equivalent to 1 min when the number of multi-epoch observations is 2. |
| Rover traveling distance between MDPO observations | 3.75 | m | The rover travels at 7.5 m/min for 0.5 min between MDPO estimations |
| Rover traveling direction | Random | deg | Heading direction is selected from three values ($+\frac{\pi}{3}, -\frac{\pi}{3}, 0$) randomly. |

**Table 3.** Satellite Orbital Parameters Used in the Simulation.

| Items | Value | Unit | Remarks |
|---|---|---|---|
| Initial Orbital Parameters of Satellite1 | | | |
| Perilune altitude | 300 | km | |
| Apolune altitude | 300 | km | |
| Inclination | 110 | deg | |
| Right Ascension of the Ascending Node | 0 | deg | |
| Argument of Perigee | 0 | deg | |
| True Anomaly | 0 | deg | |

**Table 3.** *Cont.*

| Items | Value | Unit | Remarks |
|---|---|---|---|
| Initial Orbital Parameters of Satellite2 | | | |
| Perilune altitude | 300 | km | |
| Apolune altitude | 300 | km | |
| Inclination | 110 | deg | |
| Right Ascension of the Ascending Node | 0 | deg | |
| Argument of Perigee | 0 | deg | |
| True Anomaly | −15 | deg | |

Tables 4–6 show the systematic errors used in the simulation. Table 4 summarizes the satellite orbit determination error value used in the simulation, i.e., $\Delta Along$, $\Delta Radial$ and $\Delta Cross$ defined in Equation (38). The value was chosen by adding a sufficient margin to the reference data from the LRO project [21]. Table 5 summarizes the time tag error used in the simulation, i.e., $d\tau_R$ defined in Equation (39). The value is based on the assumption that the rover clock is calibrated periodically by a navigation message every orbital period. Through the navigation message, the rover clock can be synchronized to the satellite clock with the uncertainty of the signal traveling time from the satellite to the rover and lander receivers, which was modeled as white noise in the simulation. The time tag error also contains an additive-type noise, which was modeled as a random walk noise that is reset to zero periodically. Table 6 summarizes the DEM model error value used in the simulation, i.e., $\Delta z_{R\ DEM}$, defined in Equation (41). Currently, the accuracy of the best existing DEM data in a vertical direction is about 3 m within a ±60–deg latitude and about 10 m near polar regions [17,18]. The same parameters were used in the following simulations unless otherwise mentioned.

**Table 4.** Overview of satellite orbit determination error used in the simulation.

| Items | Type | Value | Unit | Remarks |
|---|---|---|---|---|
| | $\Delta Along(t_i) = \omega_{OD-Along}(t_i) + c_{OD-Along}$ | | | |
| Satellite Orbit Determination Error in the Along direction | White Gaussian noise $\omega_{OD-Along}$ | 100.0 | m | $\omega_{OD\ t} = Value \times$ a random scalar drawn from the standard normal distribution. |
| | Bias noise $c_{OD-Along}$ | 200.0 | m | Bias $c_{OD}$ is a random number that is greater than or equal to $-Value$ and less than $Value$ |
| | $\Delta Radial(t_i) = \omega_{OD-Radial}(t_i) + c_{OD-Radial}$ | | | |
| Satellite Orbit Determination Error in the Radial direction | White Gaussian noise $\omega_{OD-Radial}$ | 10.0 | m | Same as above |
| | Bias noise $c_{OD-Radial}$ | 20.0 | m | |
| | $\Delta Cross(t_i) = \omega_{OD-Cross}(t_i) + c_{OD-Cross}$ | | | |
| Satellite Orbit Determination Error in the Cross direction | White Gaussian noise $\omega_{OD-Cross}$ | 100.0 | m | Same as above |
| | Bias noise $c_{OD-Cross}$ | 200.0 | m | |

**Table 5.** Overview of time tag error used in the simulation.

| Item | Type | Value | Unit | Remarks |
|---|---|---|---|---|
| | $d\tau_R(t_i) = \omega_{time\ tag} + x_{time\ tag}$ | | | |
| Time Tag Error | White Gaussian noise $\omega_{time\ tag}$ | 100.0 | ms | $\omega_{time\ tag\ t} = Value\times$ a random scalar drawn from the standard normal distribution. |
| | Random walk $x_{time\ tag}$ | 0.1 | ms/min | A random walk is a time series model $x_{time\ tag\ t}$ such that $x_{time\ tag\ t} = x_{time\ tag\ t-1} + \omega_t$ where $\omega_t$ is a discrete white noise series. Random walk noise is reset to zero periodically assuming orbit determination takes place every orbital period. |

**Table 6.** Overview of DEM error used in the simulation.

| Item | Type | Value | Unit | Remarks |
|---|---|---|---|---|
| | $\Delta z_R = \omega_{DEM} + c_{DEM}$ | | | |
| DEM Error | White Gaussian noise $\omega_{DEM}$ | 10.0 | m | $\omega_{DEM\ t} = Value\times$ a random scalar drawn from the standard normal distribution. |
| | Bias noise $c_{DEM}$ | 5.0 | m | Bias $c_{DEM}$ is a random number that is greater than or equal to $-Value$ and less than $Value$ |

*3.2. Simulation Results*

To secure statistical accuracy, a Monte Carlo simulation was conducted 100 times, and averaged data are presented for each specific scenario. Rover trajectory and model errors were renewed and created with every simulation.

The simulation results with different receiver observation errors $\Delta\nabla\omega$ are shown in Table 7. Figure 4 shows an example of the estimated rover trajectory overlaying the true rover trajectory when $\Delta\nabla\omega$ ($2\times\sigma_{\Delta\nabla\omega}$) is 0.4 m. Figure 5 shows the distribution of the position error between the true rover positions and the estimated rover positions of Figure 4. According to Figure 5, under the condition of the satellite orbital parameters shown in Table 3, the error distribution does not have a large anisotropy but may become more anisotropic for other cases, depending on the satellite orbital parameters. Figure 6 shows the GDOP history of Figure 4. As seen in Figure 6, the GDOP is calculated intermittently when both satellites are visible from both the rover and lander. Furthermore, the value of GDOP changes because of the geolocation of two satellites over the course of the simulation period, as well as within one consecutive observable period.

**Table 7.** Simulation results: MDPO algorithm performance evaluation.

| Receiver Observation Errors $\Delta\nabla\omega(2\times\sigma_{\Delta\nabla\omega})$ [m] | Total GDOP | Total UPE (2drms) [m] |
|---|---|---|
| 0.4 | 44.3 | 45.6 |
| 0.8 | 44.3 | 55.4 |
| 1.6 | 44.3 | 89.6 |
| 3.2 | 44.3 | 172.6 |

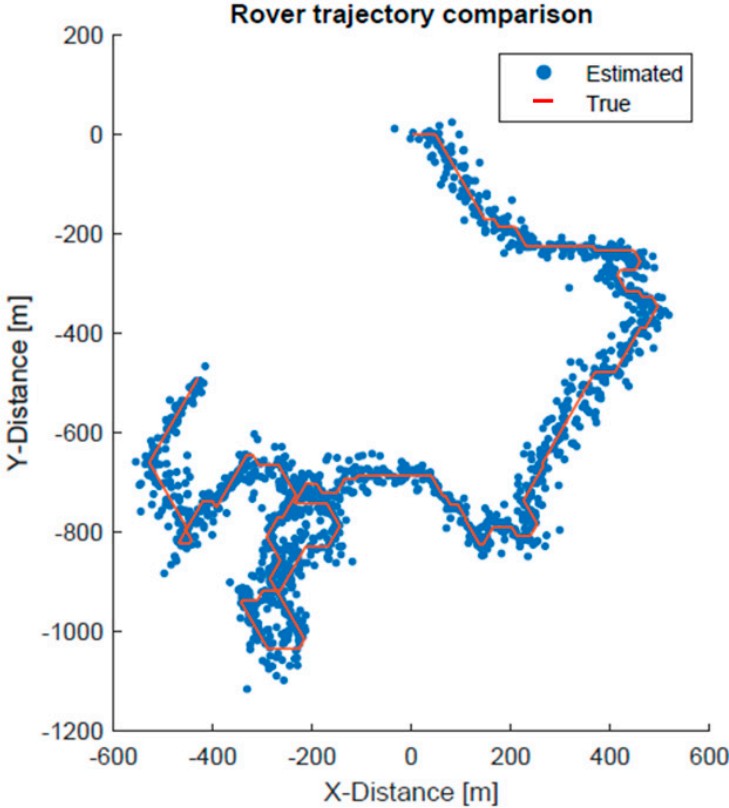

**Figure 4.** Example of simulation result: (**Red dots**) true rover trajectory; (**Blue dots**) rover positions estimated by MDPO (receiver observation errors $\Delta\nabla\omega$ ($2\times\sigma_{\Delta\nabla\omega}$) = 0.4 m).

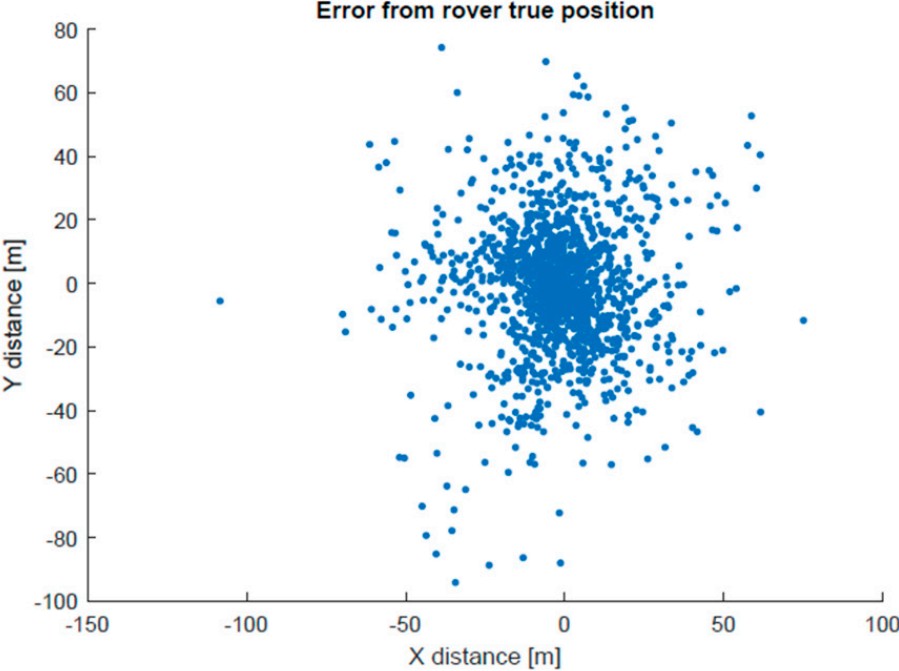

**Figure 5.** Example of position error distributions (receiver observation errors $\Delta\nabla\omega$ ($2\times\sigma_{\Delta\nabla\omega}$) = 0.4 m).

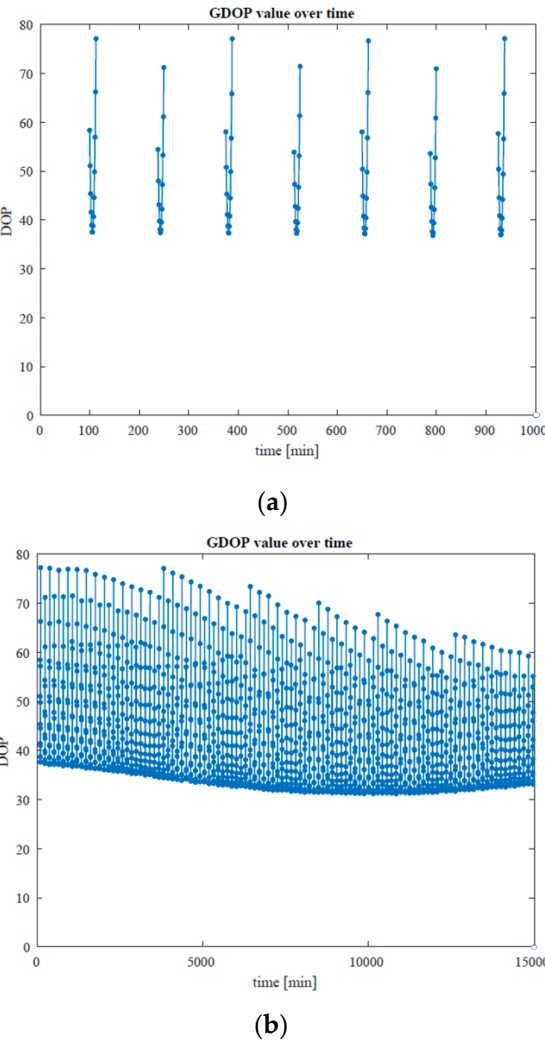

**Figure 6.** GDOP history over the course of the simulation period. (**a**) for 1000 min (closer look), (**b**) for 15,000 min (overall). Unless both satellites are in view, DOP is not calculated and not shown in the figures.

As seen in Table 7, it was confirmed that total UPE (2drms) is basically determined by the product of Total GDOP and receiver observation errors (2 $\sigma$), as indicated in Equation (30), with additional errors due to systematic errors, while the minimum UPE is determined by systematic errors when receiver observation errors are small (such as 0.4 m), as discussed in Section 2.3. It was also confirmed that the MDPO algorithm can provide a position accuracy of several tens of meters to a few hundred meters with 95% probability (2drms) within a one-minute observation, depending on the receiver observation errors.

### 3.3. Discussions

Theoretically, the MDPO algorithm works at any satellite trajectory and with any number of satellites that is more than two. Moreover, it can be evolved into a larger lunar global navigation system that is proposed by other studies [7,8], without any hardware modification. As proven by the simulation, user position error is basically determined by the product of Total GDOP and receiver observation errors according to Equation (31), while systematic errors determine a minimum user position error when the receiver observation error is small. User position error due to systematic errors is mostly derived from the steepness of the DEM function, i.e., $z_{R\ DEM}$ in Equation (38), and DEM information error, i.e., $\Delta z_{R\ DEM}$, which changes depending on the selected mission site.

## 4. Conclusions

In this research, we propose a low-cost navigation system that is based on a multi-epoch double-differenced pseudorange observations (MDPO) algorithm. MDPO requires that only two satellites be visible to locate a rover position, and unlike the conventional TOA or TDOA navigation algorithm, it also can deal with the bias of the satellite and user clocks at the same time. The numerical simulations for the considered mission scenarios demonstrated that the position error of the rover can be predicted theoretically by using the Total GDOP and receiver observation errors, with an expected exception that systematic errors induce additional user position error. It also demonstrated that the rover position can be determined within several tens of meters with a probability of 95% (2drms) within a one-minute observation using two low lunar orbits and lunar DEM information.

**Author Contributions:** Conceptualization, T.T. and S.N.; methodology, T.T.; software, T.T.; validation, T.T., T.E., and S.N.; formal analysis, T.T.; investigation, T.T. and T.E.; resources, T.T.; data curation, T.T.; writing—original draft preparation, T.T.; writing—review and editing, T.T., T.E., and S.N.; visualization, T.T.; supervision, T.T., T.E. and S.N.; project administration, T.T. and S.N.; funding acquisition, N/A. All authors have read and agreed to the published version of the manuscript.

**Funding:** This research received no external funding.

**Acknowledgments:** Yosuke Kawabada (Nakasuka-Funase laboratory at the University of Tokyo) is kindly acknowledged for his technical support in satellite orbit simulation. Keidai Iiyama (same) is kindly acknowledged for his technical advice in lunar GNSS systems.

**Conflicts of Interest:** The authors declare no conflict of interest.

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
