# Peer review of "Dual-Satellite Lunar Global Navigation System Using Multi-Epoch Double-Differenced Pseudorange Observations"

_aerospace, doi:10.3390/aerospace7090122_

Round 1

Reviewer 1 Report

Summary of Review:

This paper introduces the Multiple-Epoch Double Differencing Pseudorange Observation (MDPO) method to do positioning estimation using only two satellites in orbits. As much as this idea is somewhat interesting, the paper lacks the mathematical and analytical vigor for a peer-reviewed journal paper. There are a number of questionable assumptions and serious weakness that needed to be address. 

Comments and Suggestions: 

  1. The MDPO scheme performs range measurements in at least two epochs separated by at least one minutes. During this time, the Moon rotates, and the user position changes with respect to the Moon center frame.  Even though the user is static during the measurements, it would need to perform frame rotation operations to align the satellite, the reference, and the user positions of the two epochs.  Though the Moon rotates slowly (once every 27.3 days), at equator the distance travel in one minute at the equator is  ~270 meters.  The paper does not consider the frame rotation operation and the error introduces (e.g. Moon precessing). 
  2. The paper assumes the availability of the Digital Elevation Map (DEM), and assumes known altitude. This in turn reduces the problem into a 2-dimensional estimation problem as shown in equation 19.  This reasoning is not correct as DEM is a function of Longitude and Latitude, which are not known to start with.  The altitude error would contribute to the Dilution of Precision (DOP).  The correct problem formulation is to assume Earth center as a faux satellite, and to take into account altitude error (say Gaussian) in the error estimation and in the calculation of DOP as shown in equations 13 – 18. 
  3. The paper considers altitude estimation error in the simulation section (Section 3), yet it assumes the DEM to be a perfect sphere. Lunar DEM data with 5-m resolution (LOLA data set) are publicly available, and the authors should use the true lunar DEM data in the simulations. 
  4. The paper says in multiple places that the MDPO method is a real-time scheme. But positioning determination requires at least two epochs for measurements separated by at least 1 minute.  During this time, the user has to be idle.  As such the scheme is not real-time. 
  5. The paper uses the term “GPS” in multiple places to describe the position determination method and its error. This is a misnomer, as GPS refers to USA’s Global Positioning System. 
  6. In the abstract and in Section 2.2, the paper refers the pre-known user altitude (by knowing the DEM) as an “improvement”. This is not correct in general as the known altitude is required for 2D position determination when doing position determination in 2 epochs.  Only for 3 or more epochs that he pre-known user altitude can be regarded as an improvement. 
  7. The paper comments in multiple places that Doppler-based techniques are impractical as they would need a stable clock and/or a stable radio frequency, and that would be expensive. But lunar missions are inherently expensive, and the missions can afford Chip Scale Atomic Clock (CSAC) or even Deep Space Atomic Clock (DSAC). 

In the introduction, the paper discusses different TOA, TDOA, AOA, and FDOA schemes, which requires at least two satellites. The paper does not mention a recently introduced method by authors of reference [11] of the paper that uses a combination and range and Doppler measurements, known as Joint Doppler and Ranging (JDR), and achieve 3D positioning with as little as one satellite. 

Reviewer 2 Report

The paper concerns the possibility to use two satellites in low orbit around the Moon to provide a Navigation service.

The most important concerns for the paper are that the DD along epochs is not totally new and that the method is not compared with other ones. Moreover, the DOP problem seems not clear to the authors. More issues are listed below. 

The paper has many issues:

  • Most important: the proposed method is not new; it is widely used in satellite navigation to stack measurements from different epochs (also in case of DD measurements, see for example the Bernese or RTK manuals); moreover, generally it is also implemented a cinematic model for the rover, if I well understood all the formulas, the author didn't make any assumption so the rover should be stationary during the measurement epochs. I suggest to not claim for novelty and to state clearly that the rover is fixed during the measurement (if it is the case);
  • Important: no comparison with other methods is reported and no discussion about availability and accuracy is done, are they enough for lunar missions?
  • EQ 9-12 must be explained better they are not clear for the reader: some symbols are missing, explain that it is a linearization etc; I suggest also to put here the minimum number of equations that you need to solve the problem;
  • Spends some words on eq 13, to explain where it comes from (Jacobian etc.) and that it contains only geometrical information (you can find some details in any GNSS book;
  • eq 14 are not explained and in eq. 15 deltadeltaomega is not defined (please cite the fact that the measurement noise is improved because is the sum of the receivers measurement noise);
  • I'm not really sure that what you in par. 2.2 claim is correct: first of all the HDOP cannot be you talk only about VDOP (z-direction) and this is not correct because also the error cross-track direction of the satellites (if in the same orbit) can be very high. In general, the DOP will be very high using only two satellite that for sure will be in the line of sight (and probably the problem is bigger in cross-track directions than in z-direction).  In my opinion, it is better to say that you use the DEM model to reduce the number of unknowns.
  • Time tagging and DEM: refer some work to explain why you use these model of errors, and in particular explain why only bias is considered (no additive noise?)
  • par2.3 why calling effective Navigation Performance, usually "the percentage of time... " is called availability, please check; moreover, the availability is very low (up to 15%), can you spend some words on this in your discussion? is this enough for lunar missions?
  • Table 4. Explain why you use the random walk for the satellite error;
  • Fig 6. It is not clear how the points are correlated;

Minors:

  • line 357 rove-->rover
  • line 336 GPS?? 
  • line 226 topocentric?
  • line 161 maybe measurement-->value

Round 2

Reviewer 1 Report

Summary of Review:

The authors have adequately addressed most of my prior comments, except for a more vigorous analysis on the altitude error in the 2-dimensional estimation case (item 2 in previous review). However, since the authors also simulated the accuracy numerically taking into account the DEM error, this should provide sufficient characterization of the accuracy, so this is fine with me. I recommend that this version is ready for publication. Having said that, I would like to suggest a few areas of improvement (advisories). 

Advisories on improvements: 

  1. The whole purpose of doing multi-epoch measurements is to reduce the number of satellites required to meet a certain coverage requirement, whether it is global or regional. An interesting addition to this paper would be to compare the number of satellites required to meet a given coverage requirement by the MDPO architecture compared to the conventional GNSS approach. 
  2. The authors claimed that the use of 2-way ranging would impose extra burden and cost to the user. This is not quite true in the operation scenarios for near-Earth and deep space ranging.  In 2-way ranging, the satellite or the ground antenna sent out a ranging signal.  The user “turns-around” (or transponds) the signal back to the satellite or the ground antenna, who would then estimate the roundtrip light time.  So, the burden to the user is not high.  In fact, the TDRSS and the DSN antennas have been using 2-way ranging operationally to support Earth-orbiting and deep space spacecraft for decades.  Note that unlike GPS, 2-way ranging is a one-on-one operation that ties up one infrastructure element at a time.  But since the number of users is small, as in the lunar case, this mode of operation is adequate. 
  3. (Editorial) The paper uses the term 2drms in a few places without defining what it is.

Reviewer 2 Report

none